# Comparing the Microbiome of the Adenoids in Children with Secretory Otitis Media and Children without Middle Ear Effusion

**DOI:** 10.3390/microorganisms12081523

**Published:** 2024-07-25

**Authors:** Oļegs Sokolovs-Karijs, Monta Brīvība, Rihards Saksis, Maija Rozenberga, Laura Bunka, Francesca Girotto, Jana Osīte, Aigars Reinis, Gunta Sumeraga, Angelika Krūmiņa

**Affiliations:** 1Department of Otolaryngology, Riga Stradiņš University, 16 Dzirciema Street, LV-1007 Riga, Latvia; gunta.sumeraga@rsu.lv; 2AIWA Clinic, 241 Latgales Street, LV-1019 Riga, Latvia; 3Latvian Biomedicine Research and Study Center, 1 Ratsupites Street, LV-1067 Riga, Latvia; monta.briviba@biomed.lu.lv (M.B.); rihards.saksis@biomed.lu.lv (R.S.); maija.rozenberga@biomed.lu.lv (M.R.); laura.bunka@biomed.lu.lv (L.B.); 4Faculty of Medicine, Riga Stradiņš University, 16 Dzirciema Street, LV-1007 Riga, Latvia; 042253@rsu.edu.lv; 5“Centrālā Laboratorrija”, 1b. Šarlotes Street, LV-1011 Riga, Latvia; jana.osite@laboratorija.lv; 6Department of Biology and Microbiology, Riga Stradiņš University, 16 Dzirciema Street, LV-1007 Riga, Latvia; aigars.reinis@rsu.lv; 7Department of Infectology, Riga Stradiņš University, 16 Dzirciema Street, LV-1007 Riga, Latvia; angelika.krumina@rsu.lv

**Keywords:** adenoids, otitis media with effusion, microbiome, bacteria, 16S rRNA genetic sequencing

## Abstract

Background: The adenoids, primary sites of microbial colonization in the upper airways, can influence the development of various conditions, including otitis media with effusion (OME). Alterations in the adenoid microbiota have been implicated in the pathogenesis of such conditions. Aim: This study aims to utilize 16S rRNA genetic sequencing to identify and compare the bacterial communities on the adenoid surfaces of children with OME and children with healthy middle ears. Additionally, we seek to assess the differences in bacterial diversity between these two groups. Materials and Methods: We collected adenoid surface swabs from forty children, divided into two groups: twenty samples from children with healthy middle ears and twenty samples from children with OME. The V3-V4 hypervariable region of the bacterial 16S rRNA gene was amplified and sequenced using the Illumina MiSeq platform. Alpha and beta diversity indices were calculated, and statistical analyses were performed to identify significant differences in bacterial composition. Results: Alpha diversity analysis, using Pielou’s index, revealed significantly greater evenness in the bacterial communities on the adenoid surfaces of the healthy ear group compared with the OME group. Beta diversity analysis indicated greater variability in the microbial composition of the OME group. The most common bacterial genera in both groups were *Haemophilus*, *Fusobacterium*, *Streptococcus*, *Moraxella*, and *Peptostreptococcus.* The healthy ear group was primarily dominated by *Haemophilus* and *Streptococcus*, whereas the OME group showed higher abundance of *Fusobacterium* and *Peptostreptococcus.* Additionally, the OME group exhibited statistically significant higher levels of *Alloprevotella*, *Peptostreptococcus*, *Porphyromonas*, *Johnsonella*, *Parvimonas*, and *Bordetella* compared with the healthy ear group. Conclusion: Our study identified significant differences in the bacterial composition and diversity on the adenoid surfaces of children with healthy middle ears and those with OME. The OME group exhibited greater microbial variability and higher abundances of specific bacterial genera. These findings suggest that the adenoid surface microbiota may play a role in the pathogenesis of OME. Further research with larger sample sizes and control groups is needed to validate these results and explore potential clinical applications.

## 1. Introduction

The upper respiratory tract is a complex ecosystem hosting diverse microbial communities influenced by intrinsic and extrinsic factors such as immune responses, age, and lifestyle habits. Microbial colonization begins at birth, with significant changes occurring during the first year of life, influenced by the method of childbirth and immune system maturation. In adults, the URT microbiome becomes more diverse and stable [1]. Specific niches within the URT, such as the nasal cavity, nasopharynx, and adenoids, each host unique microbial populations that play a crucial role in maintaining health and contributing to disease development [1].

Microbial colonization in the nasopharynx, including the adenoids, is essential for the onset of infectious diseases. The pathogenesis of infections requires microorganisms to overcome host defenses and compete with resident microbiota, making it important to understand the richness and diversity of the hosts’ innate microbiome at the site of pathogen colonization [2]. Studies have shown that bacterial biofilms in inflamed adenoids can disrupt epithelial surface integrity, leading to chronic inflammation and infections [3].

The adenoids themselves are a part of Waldeyer’s ring and serve as a first line of defense by screening inhaled pathogens. The adenoid microbiome is of particular interest due to its potential role in respiratory diseases like chronic rhinosinusitis and various forms of otitis media—both acute and chronic.

Recurrent upper airway infections contribute to a severe enlargement of the adenoid tissue. These hypertrophic adenoids can obstruct the Eustachian tube, causing negative pressure in the middle ear and leading to OME [4,5].

Otitis media with effusion is defined as the presence of liquid in the middle ear without any associated signs of ear infection [6]. It is a common childhood disease, and a routine inspection of an eardrum can identify the presence of otitis media with effusion in 15% to 40% of children between the ages of one and five [7]. Most cases resolve spontaneously within 3 months; however, up to 30–40% of cases involve recurrent OME, and 5–10% episodes lasting 1 year or longer [8,9]. While some cases may resolve without intervention, the condition’s subtle symptoms of inflammation often mean it goes unnoticed for long periods. Untreated children with OME face the risk of hearing loss, which can impede language development and academic performance. Moreover, there’s a possibility of structural changes in the eardrum and middle ear, like adhesive otitis media. The preferred initial treatment is surgical insertion of tympanostomy tubes, aimed at improving hearing loss. Unfortunately, this procedure carries the potential for complications such as persistent eardrum perforation and tympanosclerosis.

Understanding this microbiome is crucial for developing effective treatments for URT diseases. Identifying the microbiota composition in adenoid tissues enhances our ability to accurately profile microbial communities and understand their impact on disease progression and treatment outcomes [2,5,10,11].

Common bacterial communities in the nose and throat of healthy individuals belong to the phyla *Firmicutes*, *Bacteroidetes*, *Actinobacteria*, and *Proteobacteria*, with prevalent genera including *Bifidobacterium*, *Corynebacterium*, *Staphylococcus*, *Streptococcus*, *Haemophilus*, *Dolosigranulum*, and *Moraxella*. The oropharynx hosts more diverse bacterial communities, with a prevalence of *Streptococcal* species, *Neisseria* spp., *Rothia* spp., and anaerobes. Detecting these microorganisms is essential for effectively treating URT infections. Various techniques have been developed over the years to isolate and research individual microorganisms on biological surfaces, with some infectious agents requiring specialized media for growth [1,6].

The development of sequencing technologies, particularly 16S rRNA gene sequencing, has greatly advanced our understanding of complex microbial communities in different body sites [7,8]. The 16S rRNA gene, notable for its conservation and specificity, provides detailed profiles of microbial diversity and serves as an ideal marker for microbial identification, allowing precise identification of bacterial taxa by targeting variable regions within the gene [10,12].

This culture-independent molecular method has allowed us to better describe adenoid bacterial composition, as well as discover previously undetected microorganisms [2,3,8]. By understanding the microbial dynamics in the adenoids, it is possible to develop targeted treatments that address specific pathogens, ultimately improving treatment outcomes [10,11,13].

## 2. Materials and Methods

### 2.1. Ethics Statement

Our research received the official approval of the Riga Stradiņš University ethics committee (approval nr. 2-PĒĶ-4/264/2022). All participants signed an informed consent. All procedures were conducted in accordance with the Helsinki Declaration on biomedical studies.

### 2.2. Patient Selection, Inclusion, and Exclusion Criteria

For our research, we used adenoid tissue collected during adenoid removal operations in children. We included children aged 3 to 7 years, both male and female. All surgeries were conducted in one location (Multifunctional outpatient clinic AIWA, Latgales Street, LV-1019 Riga, Latvia) and by one and the same certified surgeon ensuring similar techniques of operations to preserve the integrity of our results. All operations were performed under general anesthesia and in accordance with indications for adenoid removal surgeries. Operations and subsequent sample collection took place between October 2022 and October 2023. Samples were divided into two groups 20 samples each. The first group contained samples from the adenoid surface of children with no middle ear effusion, and the second group contained samples from the adenoid surface of children with otitis media with effusion at the time of operation.

#### 2.2.1. Inclusion Criteria

Patients 3 to 7 years old.Endoscopic findings—second/third/fourth-grade adenoid hyperplasia and/or computed tomography/magnetic resonance imaging findings.Tympanometry findings for the otitis media with effusion group—B type tympanogram either bilaterally or unilaterally.Tympanometry findings for the group with no middle ear effusion—A type tympanometry, strictly bilaterally.Operation under general anesthesia.Patients and their representatives agreed to participate in our research and sign an informed consent.

#### 2.2.2. Exclusion Criteria

Patients older or younger than our designated age gap.Patients and/or their representatives did not agree to participate in the study.Patients traveling to Latvia from other countries for the operation.Patients with signs of acute upper respiratory infectious disease (elevated temperature, cough, nasal purulent discharge, painful swallowing, objective signs of throat mucosal hyperemia and tonsillar exudate) as well as acute middle ear infections.Patients receiving or have received antibiotic treatment over the last two weeks.Immunocompromised patients: HIV positive/AIDS patients, hepatitis A/B/C, patients, patients undergoing chemotherapy due to oncological diseases, *diabetes mellitus* patients, patients with chronic autoimmune diseases (sarcoidosis, Wegner’s granulomatosis).Patients receiving or having received probiotics or any other microbiome-altering substances.Patients with congenital cleft malformations.

### 2.3. Material Collection

The operations were conducted under general anesthesia, so the patient was intubated first using a sterile intubation tube. The mouth and lips of the patient, as well as the oral cavity, were carefully disinfected using a chlorohexidine-based disinfection solution to prevent adenoid tissue contamination by buccal or dental microflora during extraction from the nasopharynx. The adenoids were removed using a curette with visual assistance using a nasal endoscope then were carefully removed through the mouth without touching the surface. The adenoid tissue was placed upon a sterile surface and a sterile swab was inserted into the surface and rotated several times. The swab was placed in the “Copan eNAT” system, and the sample was immediately transported to the Latvian Biomedicine Study Center for preservation and sequencing.

### 2.4. Sample Processing and Sequencing

Microbial DNA was extracted from samples using the FastDNA Spin Kit for Soil (MP Biomedicals, Santa Ana, CA, USA) following the manufacturer’s protocol. Sample lysis was performed with the FastPrep-24™ Instrument (MP Biomedicals USA, 6 Thomas, Irvine, CA 92618, USA). The V3-V4 hypervariable region of the bacterial 16S rRNA gene was targeted for amplification using the primer pair 341F/805R, primers in Table 1. During a second round of amplification, unique oligonucleotide sets were added to each sample for dual indexing. Two negative controls were included in the PCR process, checked by agarose gel electrophoresis, and sequenced with one negative control per each of the two sample batches. Post amplification, samples were purified using magnetic beads (Macherey-Nagel, Düren, Germany). The quantity and quality of the extracted DNA and amplicon libraries were assessed using the Qubit Fluorometer (Thermo Fisher Scientific, Waltham, MA, USA) and the Agilent 2100 Bioanalyzer (Agilent, Santa Clara, CA, USA), respectively. Sequencing was carried out on the MiSeq System (Illumina, San Diego, CA, USA) using the MiSeq Reagent Kit v2 (500 cycles), yielding a minimum of 100,000 paired-end reads per sample.

### 2.5. Data Analysis

The raw sequencing data were quality controlled using the “FastQC” (v0.11.9) and “MultiQC” (v1.14) software packages [14]. To increase the proportion of high-quality sequences, sequence filtering based on adapter content following the V3 and V4 hypervariable region-specific “Illumina” primer sequences was performed by excluding these sequences as sequencing artifacts. Simultaneously, sequences were trimmed to a length corresponding to an average sequence PHRED score quality of at least 20, using “cutadapt” (v4.14) [14]. “PANDAseq” (v2.11) was utilized to merge forward and reverse reads and concatenate unmergeable sequences, with a minimum overlap threshold for merging sets at 20 base pairs [15]. The prepared and merged data were denoised using the “DADA2” plugin within the “QIIME2” (v2024.2) microbiome bioinformatics platform [16,17]. To further minimize noise, resulting Amplicon Sequence Variants (ASVs) were frequency filtered for a minimum of 10 sequences per sample. A Naïve Bayes classifier integrated into the “QIIME2” environment was trained using the “SILVA” (v138.1) rRNA database to determine the taxonomic profile of the ASVs [18]. Library preparation contaminants were detected using the “decontam” (v1.18.0) Bioconductor package by comparing taxa relative frequencies in the real and negative control samples [19] according to the sequencing batches. Sequencing batch-specific decontaminated feature tables were subsequently merged and prepared for downstream statistical analysis to account for batch-specific error models. The “phyloseq” (v1.42.0) package was used on the decontaminated data to prepare a complex data object used further in the analysis [20]. Amplicon Sequence (ASV) level Inverse Simpson’s, Shannon diversity index, and Pielou’s evenness indices were calculated to evaluate both sample- and group-level alpha diversity. Group-level Shannon’s, Piellou’s, and Simpson’s index values were compared using the Wilcoxon rank sum test. ASV-level PCA was generated to evaluate intergroup differences and similarities using the “microViz” (v0.10.10) package. Statistical testing of intergroup differences and confounding variable effects was performed with a PERMANOVA test, using 99,999 permutations and a fixed seed. Finally, differential abundance tests between the experimental groups, while correcting for the sample sequencing group as a covariate, were carried out using the “Maaslin2” method with the default parameters from the “Maaslin2” (v1.16.0) Bioconductor package [21,22]. All visualizations were created using the “ggplot2” (v3.4.2) package.

## 3. Results

### 3.1. Population Statistics

A total of 40 samples were included in the study. Twenty samples were included in the first group of children with no middle ear effusion, and twenty samples were included in the group with children suffering from otitis media with effusion. There were 23 male patient samples and 17 female patient samples. The descriptive statistics based on age are shown in Table 2.

As seen in the table, the mean age is 4.45, with the Shapiro–Wilk test indicating a significant deviation from normal distribution (*p* < 0.01). For detailed age and gender distribution, refer to Table 3.

### 3.2. Alpha Diversity

We analyzed ASV level alpha diversity using the Inverse Simpson index, Shannon’s diversity index, and Piellou’s evenness index. Alpha diversity detailed in Figure 1. 

The mean Inverse Simpson index across both groups was 5.37 (SD = 4.57), indicating a high level of bacterial diversity in our samples. The Inverse Simpson index in the first group was 7.04 (SD = 5.63), and in the second group, it was 4.08 (SD = 3.27). This suggests that the bacterial communities in the adenoid tissues of children with otitis media and those with healthy ears are comparably balanced and diverse. To determine statistically significant differences in bacterial diversity in our study groups, we applied the Wilcoxon Signed-rank test. The results show no significant difference among the study groups (P.adj. = 0.13).

Our second alpha diversity test was Shannon’s diversity index. The mean result for both groups was 1.95 (SD = 0.83) for both groups, while the first group had an average index value of 2.18 (SD = 0.95), and the second group had an average index value of 1.77 (SD = 0.70). The Wilcoxon signed-rank tests once again showed no statistically significant difference in bacterial diversity between the study groups (P.ad. = 0.23).

Lastly, Piellou’s evenness index showed a mean result of 0.50 (SD = 0.19), with the first group showing an average index value of 0.57 (SD = 0.2), and the second group’s average index value was 0.45 (SD = 0.16). Wilcoxon signed-rank test was used, and it showed a statistically significant difference between the first (no ear pathologies) and the second group (P.adj. = 0.036)), indicating that the second—middle ear effusion group—has a more uneven taxonomic distribution and therefore fewer, more dominant species.

To sum up, our comprehensive analysis using Simpson’s index and Shannon’s diversity index shows no statistically significant difference in alpha diversity among the two study groups; however, Piellou’s evenness test did indicate a statistically significant difference (*p* < 0.05), resulting in the middle ear effusion group exhibiting a less even distribution of bacterial taxonomy.

### 3.3. Beta Diversity

Beta diversity is depicted in Figure 2. Aitchison distance PCA plots are divided into two subgroups representing our two study groups, also distinguished by color. In the principal component analysis (PCA) plot, we observed that the ellipse representing samples from the first group (no middle ear effusion) encompassed the ellipse of the second group (middle ear disease), indicating a spatial arrangement suggestive of greater variability within the microbial composition of the second group. A PERMANOVA test with 99,999 permutations also showed no significant difference between the variation in both groups (*p*-value = 0.46, R^2^ = 0.025).

### 3.4. Taxonomy

The phylum-level taxonomic analysis revealed a notable dominance of the Proteobacteria phylum in both groups. In the first group, proteobacteria were the most prevalent, dominating the bacterial community. Similarly, the second group also exhibited proteobacteria as the most dominant phylum, although to a lesser degree compared with the first group. The second most common phylum in both groups was Firmicutes. Interestingly, the second group (healthy ears) showed a greater relative abundance of Firmicutes compared with the first group. The third and fourth most common phyla were Fusobacteriota and Bacteroidota, respectively, in both groups. The second group exhibited a higher prevalence of both Fusobacteriota and Bacteroidota compared with the first group. Our findings are depicted in Figure 3.

In the genus-level analysis, the most identified genera in the first group were *Haemophilus*, *Streptococcus*, and *Moraxella*. Among these, *Haemophilus* was the most prevalent genus, followed by *Streptococcus* and *Moraxella*. Other notable bacteria identified in this group, though to a lesser degree, included *Fusobacterium* and *Burkholderia*. Additionally, *Parvimonas*, *Prevotella*, *Ralstonia*, *Neisseria*, and *Veillonella* were identified in minimal capacities.

The second group had a similar bacterial composition. However, the dominant genus in this group was *Fusobacterium*, with *Haemophilus* being the second most common. Additionally, *Streptococcus*, *Burkholderia*, and *Moraxella* were commonly identified in this group as well. *Peptostreptococcus*, *Alloprevotella*, and *Fusobacterium* were significantly more abundant in the second group compared with the first group. Genus-level results can be seen in Figure 4.

### 3.5. Differential Abundance

A genus-level differential abundance test between both studied groups using the Maaslin2 package with the default parameters (FDR < 0.25) revealed six statistically significant results, pointing to the involvement of these specific genera in driving the nuanced differences in the microbiome signature between them. The largest relative abundance differences were observed within the *Bordatella* genus (Coef = 4.75, p-adj. = 0.249), while the most statistically significant driver was the *Alloprevotella* genus (Coef. = 2.71, p-adj. = 0.054). Detailed results can be seen in Table 4.

## 4. Discussion

Our study comprised 40 adenoid surface swab samples, with twenty samples in each of the two groups. Although this sample size might seem limited for bacterial composition research, it is important to note that 16S rRNA studies often involve smaller sample sizes due to the high cost of analysis. Comparable studies in this field have used similar sample sizes, making our approach consistent with standard research practices [23,24,25]. We recognize that the limited sample size diminishes the statistical power of our analysis. Future studies with larger sample sizes in each group will improve the validity of our results.

Our previous work in 2023 yielded different results when comparing the adenoid microbiome between similar groups, partly due to uneven sample sizes. In our current study, we have ensured similar sample sizes in each group, enhancing the reliability and credibility of our analysis [25,26]. Despite the apparent incremental nature of changes between our publications, the current results are notably significant, particularly because we identified statistically significant differences between the study groups.

While conventional bacteriology panels could facilitate a larger sample size due to their lower costs, these methods are limited to detecting bacteria that can grow on specific agar media, which cater to particular metabolic needs. In contrast, genetic sequencing provides a comprehensive assessment of the bacterial composition, regardless of whether the bacteria can grow on the specialized agars used in traditional culturing methods.

Another point of contention is a supposition of surface and core swabs. As reported by Katundu et al., the microbiome of palatine tonsils differs between core swabs and surface swabs, with *Neisseria* and *Haemophilus* being the dominant bacterial species in the core swabs, although surface swabs contained significantly more bacterial species than the core swabs, which is logical keeping in mind that the surface of tonsils directly contacts the nonsterile mouth region [27]. A publication by Brook et al. also suggests a difference between core and surface swabs, reporting different strains of methicillin-resistant *S. aureus* in core and surface palatine tonsil swabs from the same patient [28]. Although comparing both core and surface swabs would provide a more comprehensive analysis, doubling our sample size was not feasible due to financial constraints.

We were not able to implement electronic microscopy in our assessment of bacterial biofilms due to limited access to electronic microscopes in our region. We acknowledge the significant role that biofilms in the nasopharyngeal region may play in the development of chronic diseases. For example, a comparison between the adenoids of children with chronic rhinosinusitis (CRS) and those of children who underwent surgery for obstructive sleep apnea revealed striking differences. The study found that 95% of the adenoid surface in children with CRS was covered by bacterial biofilms, compared with only 1.9% in the adenoids of patients with sleep apnea. These findings highlight the potential role of chronic inflammation in the adenoids, which may contribute to persistent infections in adjacent structures [29,30,31].

Adenotomy, a procedure aimed at improving middle ear ventilation and optimizing Eustachian tube function, has been a longstanding treatment for middle ear conditions. This surgical intervention not only facilitates better airflow but also reduces the bacterial load in the nasopharynx, potentially contributing to a lower incidence of subsequent infections [32]. Conversely, other factors may also contribute to the development of otitis media with effusion. Cheng et al. reported that variations in age, body mass index, adenoid enlargement grade, presence of allergic rhinitis, breastfeeding status, and exposure to environmental tobacco smoke had a statistically significant impact on the development of middle ear effusion, as determined through univariate and regression analyses [33]. A similar study by Yang et al. also reported a correlation between adenoid size and the presence of otitis media with effusion, noting that patients with adenoid hypertrophy had a higher body mass index compared with the control group [12]. While our study did not assess body mass index, a factor that could potentially influence microbial composition and health outcomes, it remains a pertinent consideration for future investigations. Including BMI assessments in future studies could provide additional insights into its influence on adenoid microbiota and susceptibility to OME.

Comparing our results with other author’s published works, we observe a similar bacterial pattern; however, significant differences can be identified. For example, Xu et al. investigated adenoid surface bacteria in children with otitis media with effusion. The authors report that the adenoid samples were dominated by *Haemophilus* (15.96%), *Streptococcus* (13.33%), and *Moraxella* (12.28%) [4]. Our data show a similar high abundance *of Haemophilus* and *Streptococcus*; however, we also discovered a significant colonization by *Fusobacterium*.

*Fusobacterium* on the adenoid surface was also discovered in increased relative abundance by Stapleton et al. [34]. Their method of obtaining swabs involved collecting samples before the adenoids were removed through the nasal cavity, which potentially risked contamination. In contrast, our approach involved collecting swabs only after the adenoids were removed, thereby minimizing the risk of contamination. We believe our way is much better, although it requires a surgical operation. Transnasal nasopharyngeal swab collection on the other hand can be performed outside the surgical ward without the need for a surgical intervention.

Uhliarova et al. reported a significant decrease in nasopharyngeal pathogens post adeonotmy, as well as an increase in allergy-associated diseases in children suffering from adenoid hypertrophy [35]. Allergological testing following adenoidectomy was not conducted in the current study. However, investigating the incidence and impact of allergological diseases both pre and post adenoidectomy could be a valuable area for future research. Allergies and immune responses are known to play significant roles in upper respiratory tract infections and middle ear diseases, including otitis media with effusion [36]. Understanding how adenoidectomy influences allergological profiles could provide deeper insights into the interplay between immune function and adenoid microbiota dynamics.

*Haemophilus*, as the dominant bacterial genus, has been reported by several authors [28,37,38,39]. *Haemophilus* species are pathogenic or opportunistic bacteria frequently colonizing the mucosa of the upper respiratory tract. Several serotypes of *Haemophilus* have been identified (types a through f), with type b (HiB) being the primary serotype targeted in studies, vaccination, and therapy. The introduction of HiB vaccines has significantly reduced the prevalence of this serotype [40]. In recent years, other serotypes (mostly a,e,f), as well as nontypable *H. influensae*, started to play a more significant role in the development of recurrent upper respiratory tract infections [41,42]. It is reported that up to 96% of children have nontypeable *H.influensae* colonization due to vaccination [43]. These frequent and abundant *H. influensae* findings, as well as our data, suggest a significant role of *H. influensae* in developing adenoid hypertrophy through chronic carriage and inflammation. To comprehensively assess the distribution of *H. influensae* serotypes within the population and to evaluate the impact of vaccination strategies in Latvia, more precise data on the prevalence and coverage of *H. influensae* type vaccines are essential. Unfortunately obtaining such data is currently challenging due to limitations in accessing the national vaccination database. Improved access to comprehensive vaccination records would facilitate a clearer understanding of vaccine effectiveness and the epidemiological landscape of *H. influensae* infections in the region. It is important to note that the 16S rRNA sequencing methods used in our study do not allow for differentiation between *H. influensae* serotypes, which highlights the need for complementary typing methods in future investigations.

Other notable bacterial species that colonize the adenoid surface are *Streptococci*. Our data show a higher abundance of *Streptococcus* genus on the surface of the adenoids in children with healthy compared with children with secretory otitis media. This finding contrasts with Bernstein et al., who reported a lower prevalence of Streptococcus viridans in children with recurrent middle ear effusion [44]. The findings may be influenced by advancements in technology since that time, particularly in the field of microbial identification techniques, notably improvements in sequencing technologies and bioinformatics tools. It is plausible that discrepancies in findings between studies, including ours and those conducted decades ago, may partially reflect these technological advancements and their impact on *Streptococcus* identification and microbial profiling.

We identified several adenoidal bacteria in the otitis media with effusion group that have a statistically significant presence in the microbiome, notably *Alloprevotella* and *Peptostreptococcus*.

*Alloprevotella* is linked to dental caries, gum disease, periodontitis and, in severe conditions, mandibular osteomyelitis [45,46]. The greasy coating of the tongue, containing multiple bacterial species, predominantly *Alloprevotella* and *Streptococcus*, is also considered an early biomarker of gastrointestinal malignancy [47]. *Alloprevotella* migration towards the nasopharynx potentially serves as a dual risk factor: it may contribute to eustachian tube dysfunction and subsequent accumulation of middle ear effusion, particularly given its association with severe dental disorders.

*Peptostreptococcus* was identified in a notable abundance in the middle ear effusion group but was almost not present in the healthy middle ear group. *Peptostreptococcus* is hard to isolate due to its long growth period on conventional agars. Advances in molecular genetics and genetics-based bacterial identification allow us to identify this bacterial genus more frequently and with more precision. *Peptostreptococcus* genus bacteria are linked to periodontal infections, as well as a marker for colorectal cancer [48,49]. *Peptostreptococcal* invasion and hematogenous dissemination may also cause septic arthritis [50]. While our study primarily focused on the pediatric population and did not specifically investigate conditions like septic arthritis or colorectal cancer associated with *Peptostreptococcus*, it may also play a role in influencing middle ear pathologies.

Scalfani et al. recently investigated the role of excessive bacterial colonization in chronic inflammation leading to adenoid hypertrophy. Their study is based on the hypothesis that reducing the bacterial load in the nasopharynx with an antibacterial agent—specifically, a 30-day course of amoxicillin and clavulanic acid—might mitigate the severity of conditions associated with adenoid hypertrophy. The authors reported that this treatment significantly reduced the need for surgical intervention in children with obstructive adenotonsillar hypertrophy, as observed at a 1-month follow-up [51]. Antibiotic use prior to adenoidectomy was an exclusion criterion in our study. Prolonged use of antibiotics for recurrent upper airway diseases is generally discouraged in Latvia. However, findings such as those by Scalfani may prompt some specialists to consider extended antibiotic regimens in specific cases.

An intriguing study by Jesic et al. found reduced alkaline phosphatase activity in adenoid tissue in children with otitis media with effusion compared with those with no middle ear effusion. The authors noted that a definitive explanation for this finding has yet to be determined [52]. The identification of alkaline phosphatase activity is not typically part of routine investigations, and we currently lack the technical methods necessary to validate the authors’ findings.

Another significant topic of discussion is the evolution of the adenotonsillar microbiome over time, particularly as associated diseases progress or resolve. At present, there are no studies that track changes in bacterial composition in the adenoids throughout a patient’s lifetime [53]. An increase in symptom severity may reflect changes in bacterial colonies, several species may assume a more dominant role, and other species may not be detected altogether. If the opportunity arises, we would like to continue our research in this field as well.

In the contemporary literature, studies examining the adenoid microbiome often do not distinguish between children with otitis media with effusion, obstructive sleep apnea, or other adenotonsillar-disease-associated conditions [54,55]. Our study highlights that the adenoid microbiome is not static and exhibits variability across different pathological conditions. Specifically, we demonstrate distinct microbial compositions between children with OME and those without middle ear effusion, emphasizing the need for nuanced investigations into how microbial profiles may influence disease pathogenesis and treatment outcomes. By identifying predominant bacterial genera such *as Haemophilus*, *Streptococcus*, *Fusobacterium*, and *Peptostreptococcus* in these populations, we underscore their potential roles in the pathogenesis of OME. Understanding these microbial profiles can potentially inform targeted treatment strategies, including antibiotic selection and the development of microbiome-based therapies, to mitigate the burden of OME. Our findings may pave the way for personalized medicine approaches tailored to the microbial profiles of individual patients, ultimately optimizing treatment outcomes in pediatric otolaryngology.

## 5. Conclusions

By comparing the bacterial colonies on the adenoids of children with healthy middle ears and children with otitis media with effusion, we identified statistically significant differences in alpha diversity using Pielou’s index, with the healthy middle ear group showing greater evenness. Beta diversity analysis indicated greater variability within the microbial composition of the middle ear effusion group compared with the healthy middle ear group.

The most common bacterial genera in both groups were *Haemophilus*, *Fusobacterium*, *Streptococcus*, *Moraxella*, and *Peptostreptococcus*. The healthy ear group was dominated by *Haemophilus* and *Streptococcus*, while the middle ear effusion group had an abundance of *Fusobacterium* and *Peptostreptococcus*. Additionally, we found statistically significant differences in the abundance of several genera in the middle ear effusion group, with higher levels of *Alloprevotella*, *Peptostreptococcus*, *Porphyromonas*, *Johnsonella*, *Parvimonas*, and *Bordetella* compared with the healthy ear group.

In future studies, we would like to increase the sample size and include a control group of completely healthy children. However, obtaining nasopharyngeal swabs from healthy children without causing discomfort and avoiding sample contamination may present challenges.

## Figures and Tables

**Figure 1 microorganisms-12-01523-f001:**
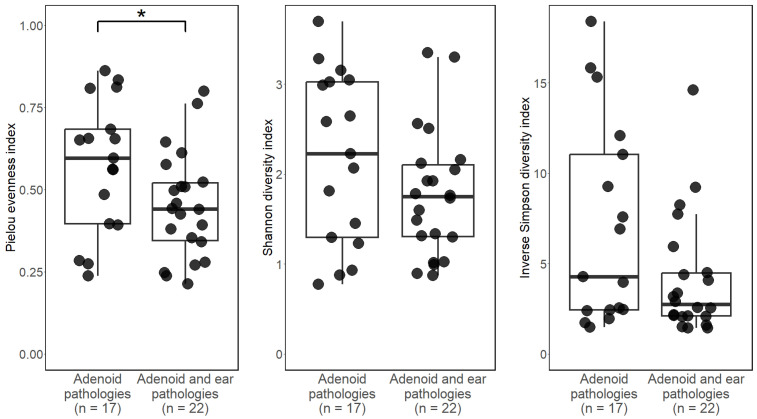
Shannon’s diversity index, Simpson’s diversity index, and Piellou’s evenness test. Boxplots of median values and interquartile values for the two study groups. No statistically significant difference in bacterial diversity between the two study groups was identified using Wilcoxon signed-rank tests (*p*-value > 0.05) in Simpson’s and Shannons’s indexes. Statistically significant differences were identified in the Piellou evenness index values (*p*-adjusted—0.036), also indicated by the (*) sign above the boxplots.

**Figure 2 microorganisms-12-01523-f002:**
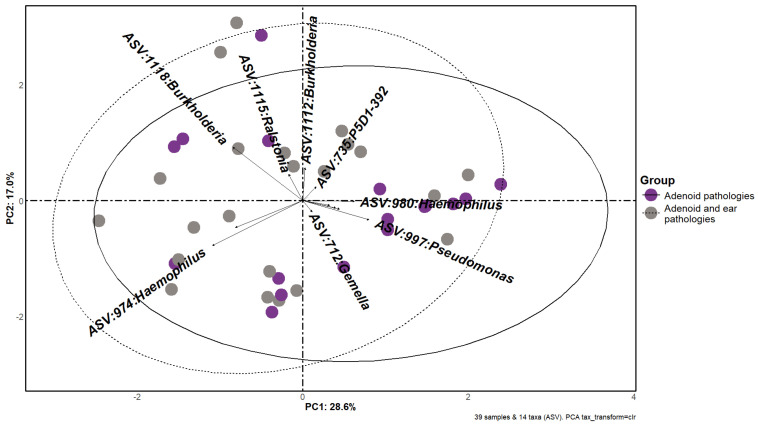
Beta diversity plot shows a comparison between the two study groups. Different colors represent different subgroups (patients with middle ear effusion and without middle ear effusion) and two sequencing batches in triangle and circle form. Separation of samples is based on principal components, and the t multivariate distribution ellipse confidence level is set to 95%. We observe the first group’s ellipse almost completely encompassing the second group’s ellipse, signaling high intragroup heterogeneity in both groups.

**Figure 3 microorganisms-12-01523-f003:**
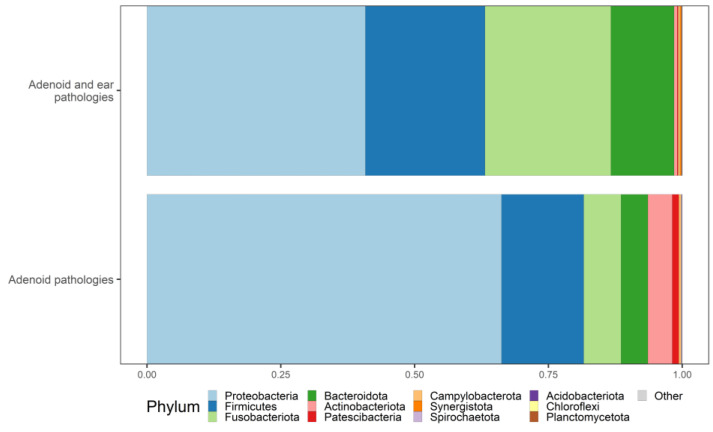
Taxonomy bar plot illustrating the relative abundances of bacterial phyla. Samples are grouped and sorted according to their respective comparison groups.

**Figure 4 microorganisms-12-01523-f004:**
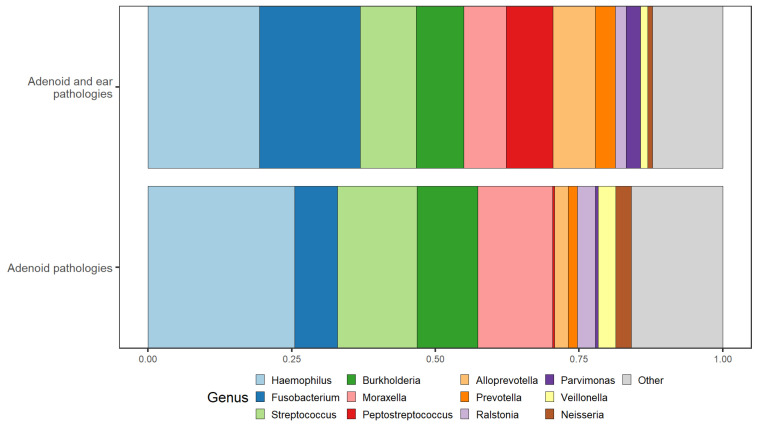
Taxonomy bar plot illustrating the relative abundances of bacterial genera. Samples are grouped and sorted according to their respective comparison groups.

**Table 1 microorganisms-12-01523-t001:** V3-V4 primer sequences.

Name	Sequence
ci5_16S_V3_Fw(341F)	TCGTCGGCAGCGTCAGATGTGTATAAGAGACAGNNNNNNCCTACGGGNGGCWGCAG
ci7_16S_V4_Rs(805R)	GTCTCGTGGGCTCGGAGATGTGTATAAGAGACAGNNNNNNGACTACHVGGGTATCTAATCC

**Table 2 microorganisms-12-01523-t002:** Descriptive statistics.

	N	Mean	SD	Shapiro–Wilk	S-W *p*-Value
Age	40	4.45	1.45	0.837	<0.01

**Table 3 microorganisms-12-01523-t003:** Distribution by age in our study groups.

	Group 1—No Middle Ear Effusion	Group 2—Secretory Otitis Media
Gender	Male—15 participants	Male—8 participants
	Female—5 participants	Female—12 participants
Age	3 years old—6 participants	3 years old—8 participants
	4 years old—5 participants	4 years old—5 participants
	5 years old—3 participants	5 years old—3 participants
	6 years old—2 participants	6 years old—2 participants
	7 years old—4 participants	7 years old—2 participants

**Table 4 microorganisms-12-01523-t004:** The table presents the statistical analysis comparing the abundance of bacterial genera between the two groups. Coefficient represents the difference in abundance of a particular genus between the groups. Positive coefficients indicate higher abundance in the second group (middle ear effusion group), while negative coefficients indicate higher abundance in the first group. *p*-value less than 0.05 considered statistically significant. *p*-value adjusted for multiple comparisons to control for the false discovery rate, with lower values indicating a significant result after adjustment.

Genus	Group	Coefficient	Standard Error	*p*-Value	*p*-Value (adj.)
Alloprevotella	Middle ear effusion	2.71	0.82	9.61 × 10^−4^	0.0543
Peptostreptococcus	Middle ear effusion	3.53	1.13	1.80 × 10^−3^	0.054
Porphyromonas	Middle ear effusion	1.77	0.70	1.12 × 10^−2^	0.168
Johnsonella	Middle ear effusion	4.43	1.71	9.58 × 10^−3^	0.168
Parvimonas	Middle ear effusion	2.94	1.31	2.49 × 10^−2^	0.249
Bordetella	Middle ear effusion	4.75	2.10	2.35 × 10^−2^	0.249

## Data Availability

The raw data supporting the conclusions of this article will be made available by the authors on request.

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
