# Peer review of "Comparing the Microbiome of the Adenoids in Children with Secretory Otitis Media and Children without Middle Ear Effusion"

_microorganisms, 2024, doi:10.3390/microorganisms12081523_

Round 1
Reviewer 1 Report
Comments and Suggestions for Authors
In the article “Comparing the Microbiome of the Adenoids in Children with Secretory Otitis Media and Children without Middle Ear Effusion,” the authors employ 16S rRNA sequencing to identify and compare the bacterial communities present on the adenoid surfaces of children with otitis media with effusion (OME) and children with healthy middle ears. However, the article has several issues that need to be addressed:
Sample Size: The study uses a relatively small sample size, with two groups of 20 samples each. The authors justify this limitation by citing the high cost of 16S rRNA sequencing. While financial constraints are understandable, it is worth noting that other studies have used larger sample sizes (e.g., in https://journals.asm.org/doi/10.1128/msystems.00056-21 73 effusion samples were sequenced). This discrepancy raises concerns about the statistical power and generalizability of the findings.
Novelty and Originality: The novelty and originality of this work are questionable given that the same research group has recently published two similar studies:
https://www.mdpi.com/1648-9144/58/7/920
https://www.mdpi.com/2076-2607/11/8/1955
Particularly, the latter study also investigated the bacterial composition on the adenoid surfaces of children with OME and children with healthy middle ears using 16S rRNA sequencing. This overlap in research raises concerns about the incremental contribution of the current study.
Definitions: There is no clear definition of otitis media with effusion (OME) or secretory otitis media in the article. It is essential to explain that these terms are synonymous for readers who may not be familiar with the terminology.
Clarity and Readability: The manuscript would benefit from professional editing. Several sentences are poorly constructed, making it difficult to understand the intended message.
Discussion Section: The discussion section predominantly describes past studies without adequately correlating their findings with the current study’s conclusions. A more critical and integrative discussion would enhance the value of the article.
Considering these significant issues, I do not recommend this paper for publication in its current form.
Comments on the Quality of English LanguageThe manuscript would benefit from professional editing. Several sentences are poorly constructed, making it difficult to understand the intended message.
Author Response
To the esteemed reviewer
We are extremely grateful to you for dedicating your time and providing insightful feedback on our manuscript.Here are ways we improved our manuscript based on your valuable suggestions:
Sample Size: The study uses a relatively small sample size, with two groups of 20 samples each. The authors justify this limitation by citing the high cost of 16S rRNA sequencing. While financial constraints are understandable, it is worth noting that other studies have used larger sample sizes (e.g., in https://journals.asm.org/doi/10.1128/msystems.00056-21 73 effusion samples were sequenced). This discrepancy raises concerns about the statistical power and generalizability of the findings.
We also acknowledge the limited sample size, and a relative lack of statistical power, unfortunately, due to monetary constrains this is the best we could do in our circumstances.
However, we disagree that other publication have a substantially larger sample size. For example – this publication has a similar sample size to our own https://www.tandfonline.com/doi/full/10.1080/20002297.2023.2182571, and the authors claim a 90% strength rate of their statistical analysis.
This publication also has a similar sample size (45 samples) - https://www.mdpi.com/2076-2607/11/2/422
This publication has 20 samples less than ours including only 20 samples –
https://advancedotology.org//en/the-adenoid-microbiome-in-recurrent-acute-otitis-media-and-obstructive-sleep-apnea-131135
Honoring your correct remark regarding the strength of our statistical analysis we added a line in our discussion section where we acknowledge that a larger sample size may improve our analysis believability (Line 378)
Novelty and Originality: The novelty and originality of this work are questionable given that the same research group has recently published two similar studies:
We absolutely agree that there is significant portion of our current study overlaps with our previous published material. This is a continuation of our work started in 2022. We believe that with this current project we fixed one of the main issues we had in our previous publication, namely – the uneven group distribution.
Our current results are also completely different from our previous publications where no differences between our chosen study groups were identified. In our current work significant differences were found comparing our, currently even, study groups leading to results that may be interesting to your journals readers.
We also added our clarification regarding the seemingly incremental differences in our manuscript (line 381-387)
Definitions: There is no clear definition of otitis media with effusion (OME) or secretory otitis media in the article. It is essential to explain that these terms are synonymous for readers who may not be familiar with the terminology.
Thank you for pointing this oversight. We added a definition of otitis media with effusion and epidemiological data in the introduction section (line 68 – 80)
Clarity and Readability: The manuscript would benefit from professional editing. Several sentences are poorly constructed, making it difficult to understand the intended message.
We edited our manuscript with the help of a native English speaker as well as ran our manuscript through linguistic software which improved clarity
Discussion Section: The discussion section predominantly describes past studies without adequately correlating their findings with the current study’s conclusions. A more critical and integrative discussion would enhance the value of the article.
Thank you for pointing out our missteps in the discussion section, we have re-edited our discussion section with additional information and more connection to our results as you suggested, hopefully our discussion section is more cohesive now.
We hope our corrections in the manuscript are suitable enough to allow our work to be published, once again thank you for your valuable insight!
The authors of the manuscript
Reviewer 2 Report
Comments and Suggestions for Authors
The article compares the microbiome of edenoids in children with secretory otitis media and children without middle ear effusion. The aim of the article is quite courageous, as adenoids microbiome is known to be vey large. As the treatment could be modified by your/future research, please emphasize your results. A paragraph with treatment options/chalenges could be added and how the research could change the paradigm It remains a gap about the other factors involving local inflamation. Please add a paragraph about local inflamation process (that could also be involved in your results) The article presents a large comparison in microbiome, comparing to other articles. A brief presentation of other results/a meta-analisys could be also presented in discussion. The methods are properly used, no sugestions are to be made The conclusions are strongly related to the results, but a better soundness could be achieved with specific results (bacterial genera e.g.) The references are well chossen English grammar: no important changes are to be done, as the english language is comprehensive and properly used. Figures and tables- number, format- suitable for publication, no important duplicity in data presentation. Length of the manuscript- no inappropriate citation or recurrent data/information was found
A good article with good conclusions about microbiome of the adenoids.
Good english.
Please emphasize more the role/aim of this paper in current literature, in pathology and how it could change the treatment. The originality of the paper needs to be emphasized at this point. The metodology is well chosen, the statistics are well organised and interpreted. The conclusions are related to the aim/article manuscript. Good references chosen.
Author Response
Review 2
We sincerely appreciate the time and effort you have dedicated to reviewing our manuscript. In this letter, we address each of your comments and suggestions, providing explanations and outlining the revisions we have made.
Please emphasize more the role/aim of this paper in current literature, in pathology and how it could change the treatment
As requested, we added a paragraph to our discussion emphasizing the role of our study and impact on treatment (line 522-536)
Once again, we sincerely appreciate the time and effort you have devoted to reviewing our manuscript.Hopefully, the changes we made in our manuscript based on your insight will improve our work
The authors of the manuscript
Round 2
Reviewer 1 Report
Comments and Suggestions for Authors
I do not consider that the revised manuscript was significantly improved. There are still poorly constructed sentences and typos, and it does not seem that the manuscript was edited with the help of a native English speaker. Additionally, there is duplicated information and other issues that detract from the readability of the manuscript.
Some examples:
"Specific niches within the URT, such as the nasal cavity, nasopharynx, and adenoids, each host unique microbial populations which play a crucial role in maintaining health and contributing to disease [1]."
"However, hypertrophic adenoids can obstruct the Eustachian tube, causing negative pressure in the middle ear and leading to OME [8, 9]." / "The enlargement of adenoid tissue can block the nasopharyngeal orifice of the Eustachian tube, leading to negative pressure and mucosal changes [8, 10]."
"Samples were divided into two groups 20 samples each. The first group contained samples form adenoid surface in children with no middle ear effusion, the second groups contained samples from the adenoid surface in children suffering from otitis media with effusion at the time of operation".
"Accurate microbiological swabs of the adenoid surface can only be collected after the adenoid tissue is removed from the nasopharyngeal region. The operations are conducted under general anesthesia, so the patient is intubated first using a sterile intubation tube. The mouth and lips of the patient as well as the oral cavity is carefully disinfected using a chlorohexidine-based disinfection solution to prevent adenoid tissue contamination by buccal or dental microflora during extraction from the nasopharynx. The adenoids are removed using a curette with visual assistance using a nasal endoscope then carefully removed through the mouth without touching the surface. The adenoid tissue is place upon a sterile surface and a sterile swab is inserted into the surface and rotated several times. The swab is placed in “Copan-Enat” system, the sample is immediately transported to Latvian Biomedicine study center for preservation and sequencing" -> This sentence is in the present tense, but it should be in the past tense like the other methods described.
"H. influensae"
"Excessive bacterial colonization as a cause of chronic inflammation, that results in hypertrophy of adenoid tissue was recently investigated by Scalfani et al. Based on the theory that reducing the bacterial load in the nasopharynx using an anti-bacterial sub-stance, in this case -amoxicillin and clavulanic acid for prolonged periods of time may impact the severity of adenoid hypertrophy-associated conditions. Authors report that s 30 day course of amoxcicillin-clavulanate significantly reduced surgery in children with obstructive adenotonsillar hypertrophy at 1-month follow-up [54].
"In the future studies We should like to increase the sample size and include a control group of completely healthy children".
Additionally, the discussion section remains very descriptive and lacks critical analysis. Because of this, along with the other issues mentioned, I still do not recommend this paper for publication.
Comments on the Quality of English LanguageAs mentioned above, there are still poorly constructed sentences and typos, and it does not seem that the manuscript was edited with the help of a native English speaker.
Author Response
To the esteemed reviewer!
We would like to thank you once again for your contribution in providing valuable insight into our manuscript
We have made changes throughout the manuscript to improve grammatical mistakes and typos, especially in the places you pointed out.
Here is the list of changes in our manuscript:
Line 53 – changed to influenced by the method of childbirth
Line 57 - health and contributing to disease development
Line 60 - The pathogenesis of infections requires microorganisms
Line 65 - The adenoids themselves are a part of Waldeyer's
Line 67 - various forms of otitis media – both acute and chronic.
Line 72 - Line 84 has been completely re-written
Line 85-88 - trimmed down,
Line 86 - Identifying the microbiota composition in adenoid
Line 146- Line 151 sentences switched places and trimmed
Line 152 - Line 153 - This culture-independent molecular method has allowed us to better describe adenoid bacterial composition as well as discover previously undetected microorganisms
Line 155 - ultimately improving treatment outcomes
Line 177 - Line 178 - added missing from
Line 253 - Patients receiving or have received
Line 264 - Line 274 - adjusted to past tense
Line 473- Line 488 - re-arranged and re-written to improve fluency
Line 501- Line 503 - re-arranged and re-written to improve fluency
Line 505 - Line 513 - re-arranged and re-written to improve fluency
Line 514- Line 529 - re-arranged and re-written to improve fluency
Line 626 - Line 632 - - re-arranged and re-written to improve fluency
Line 643- Line 662 - Haemophilus italicized
Line 638 - microbiome, notably – Alloprevotella and Peptostreptococcus.
Line 751 - and haematogenous (corrected spelling)
Line 755 - Line 771 - re-written to improve fluency
Line 776 - may assume a more dominant (corrected spelling)
Line 779 - 889 - re-written to improve fluency
As well as multiple other changes to correct spelling and grammar
We value your critical opinion on our discussion section; however, we do not completely understand what exactly you mean by the “lack of critical analysis” and our text being “descriptive”.
We addressed the issues we had while developing our research and pointed out our shortcomings. We also compared our results with previously published works.
Our discussion section is also quite like the newly published article from your journal - https://www.mdpi.com/2076-2607/12/7/1483, in which the discussion section mostly touches on other authors’ works while only slightly comparing their results to the results the authors of that publication report.
We hope that the changes in our manuscript will be sufficient to allow the manuscript to proceed.
Best wishes,
The authors of the manuscript